# MAgPIE 4 - A modular open source framework for modeling global land-systems

Jan Philipp Dietrich[1], Benjamin Leon Bodirsky[1], Florian Humpenöder[1], Isabelle Weindl[1], Miodrag Stevanović[1], Kristine Karstens[1], Ulrich Kreidenweis[2], Xiaoxi Wang[1], Abhijeet Mishra[1], David Klein[1], Geanderson Ambrósio[3,1], Ewerton Araujo[4,1], Amsalu Woldie Yalew[1], Lavinia Baumstark[1], Stephen Wirth[1], Anastasis Giannousakis[1], Felicitas Beier[1], David Meng-Chuen Chen[1], Hermann Lotze-Campen[1,5], and Alexander Popp[1]

[1]Potsdam Institute for Climate Impact Research (PIK), Member of the Leibniz Association, P.O. Box 60 12 03, D-14412 Potsdam, Germany

[2]Leibniz Institute for Agricultural Engineering and Bioeconomy (ATB), Member of the Leibniz Association, Max-Eyth-Allee 100, D-14469 Potsdam, Germany

[3]Universidade Federal de Viçosa, Departamento de Economia Rural - DER, Av. Purdue s/n°, Campus Universitário, CEP 36570-900 Viçosa, MG, Brazil

[4]Universidade Federal de Pernambuco, Programa de Pós-Graduação em Economia - PIMES, Av. dos Economistas s/n°, Centro de Ciências Sociais Aplicadas, Cidade Universitária, CEP 50670-901 Recife, PE, Brazil

[5]Humboldt-Universität zu Berlin, Department of Agricultural Economics, Unter den Linden 6, 10099 Berlin, Germany

*Correspondence to:* Jan Philipp Dietrich (dietrich@pik-potsdam.de) and Alexander Popp (popp@pik-potsdam.de)

**Abstract.** The open source modeling framework MAgPIE combines economic and biophysical approaches to simulate spatially-explicit global scenarios of landuse within the 21st century and the respective interactions with the environment. Besides various other projects, it was used to simulate marker scenarios of the Shared Socio-economic Pathways (SSPs) and contributed substantially to multiple IPCC assessments. However, with growing scope and detail, the non-linear model has become increasingly complex, computationally intensive, and intransparent, requiring structured approaches to improve the development and evaluation of the model.

Here we provide an overview on version 4 of MAgPIE, and how it addresses these issues of increasing complexity using new technical features: modular structure with exchangeable module implementations, flexible spatial resolution, in-code documentation, automatized code-checking, model/output evaluation, and open accessibility. Application examples provide insights into model evaluation, modular flexibility and region-specific analysis approaches. While this paper is focused on the general framework as such, the publication is accompanied by a detailed model documentation describing contents and equations, and by model evaluation documents giving insights into model performance for a broad range of variables.

With the open source release of the MAgPIE 4 framework we hope to contribute to more transparent, reproducible and collaborative research in the field. Due to its modularity and spatial flexibility it should provide a basis for a broad range of land-related research with economic or biophysical, global or regional focus.

# 1 Introduction

Global land-use is expected to undergo major changes over the coming decades caused by population growth, climate change, climate change mitigation and various other socio-economic changes. Climate change has already had significant impacts on crop yields (Lobell et al., 2011; Rosenzweig et al., 2014), water availability (Strzepek and Boehlert, 2010) and biodiver-
sity distribution (Foden et al., 2013). Mitigation of climate change could entail large repercussions on the land-use system (Popp et al., 2017) by implementing strategies such as bioenergy mandates (Humpenöder et al., 2018), afforestation policies (Humpenöder et al., 2014) or induced changes in dietary habits (Stevanović et al., 2017). The land-use sector is also affected by the prospects of demographic and economic changes, including the increase in demand for agricultural products (Alexandratos and Bruinsma, 2012; Bodirsky et al., 2015). Finally, the global political discourse framed by the Sustainable Development
Goals (SDGs) (United Nations, 2015) will most likely cause further transformations of the land-use sector (Humpenöder et al., 2018; Pradhan et al., 2017).

In light of these challenges, methodological tools that quantify and analyze such effects and inform decision makers are required. To this end, models such as GCAM (Wise et al., 2014), AIM (Fujimori et al., 2017), GLOBIOM (Havlík et al., 2014; Kindermann et al., 2006), IMAGE (Stehfest et al., 2014), MAgPIE (Lotze-Campen et al., 2008) and others are being
developed. They combine biophysical (e.g. plant growth, land availability, water cycles) and economic (e.g. trade, production costs, policies) aspects and can be applied to a broad set of questions. Driven by the motivation to comprehensively represent many interactions and consequences of land-use and land related processes, these models have become more detailed and complex over time. Moreover, the range of questions and applications has become wider. These advancements come with the burden of increased computational requirements and increased challenges in manageability and transparency. New approaches
are required to make models more manageable, efficient and open.

This paper presents the MAgPIE 4 (Model of Agricultural Production and its Impact on the Environment 4) modeling framework which has been built to cope with the aforementioned challenges of complexity, manageability and transparency. The framework addresses these challenges via two conceptual foundations it rests on: modularity and flexibility in the level of detail.

Modularity denotes the concept of building a model as a network of separate modules reflecting its different components, instead of handling the model as a whole. A module can have different realizations, each of which gives a different representation of the sub-system it models. Building the model as a network of modules eases the understanding of the model as well as the modification of components of it.

Flexibility in the level of detail means adjusting the temporal and spatial resolution. It also means that module realizations
can be chosen based on the research question and thereby adjusting the model complexity appropriately.

The flexibility and the modular concept enable a tailor-made set-up of simulations consistent with the spatial, temporal and contextual scope of the analysis. It allows for reducing complexity where it is not needed and increasing simulation detail where it makes a difference. The resulting indefiniteness in model specification is reflected by a shift in terminology from

*model* (MAgPIE before version 4) to *framework* (MAgPIE 4 and beyond), reflecting that very different *models* of the land-use sector can be built with the same *framework*.

In the subsequent sections, we present the concept of the modeling framework MAgPIE 4 starting with a brief description of the model history, the new features in version 4 and a short overview of the modules in version 4. This is followed by a methodological section about the modeling framework explaining its technical properties such as modularity and spatial flexibility. The main text is completed by an output section – showing some specific use case of the modular structure and spatial flexibility provided by the framework – as well as a discussions and conclusions section. Supplementary material provides model code, model documentation and extended evaluation information to better embed the presented work.

## 2  Model Features

### 2.1  A brief history of MAgPIE

MAgPIE was first introduced in Lotze-Campen et al. (2008) as recursive dynamic cost minimization model, simulating crop production, land-use patterns, and water use for irrigation in a spatial resolution of three by three degrees and inter-regional trade between 10 world regions. Spatially explicit biophysical information was derived by a link to the global gridded crop and hydrology model LPJmL (Bondeau et al., 2007). Prices are implicitly modeled as marginals of the model constraints. Intensification as well as other decisions in the model are coming from an interplay of physical constraints and costs associated to activities in the model. While not being versioned at the time of publication this variant is ex-post referred to as **version 1**. Follow-up publications based on version 1 introduced different categories of unmanaged land such as undistHerbed natural forests (Krause et al., 2009, 2013). Intra-regional transport costs accounting for the travel distance to the nearest market were also introduced to this version (Krause et al., 2013). Further additions included bioenergy production (Lotze-Campen et al., 2010), $CO_2$ emissions from land-use change (Popp et al., 2012), and agricultural non-$CO_2$ greenhouse gases (Popp et al., 2010, 2011b). Moreover, this early version of MAgPIE was already coupled to an energy-system model by exchanging price and demand information on bioenergy, thereby establishing the integrated assessment modeling framework REMIND-MAgPIE (Popp et al., 2011a).

**Version 2** of the model was the first step towards spatial flexibility. The spatial 3 by 3 degree cells were replaced by clusters, which are aggregates of spatial 0.5 by 0.5 degree grid cells with similar properties. Moving from cells to clusters improved both accuracy and model performance at the same time (Dietrich et al., 2013).

In terms of content, version 2 introduced endogenous yield increases through investments into research and development (Dietrich et al., 2014), a more detailed estimation of food demand (Bodirsky et al., 2012, 2015) and marginal abatement cost curves (MACC) to model technical greenhouse gas (GHG) emission abatement (Popp et al., 2010; Lucas et al., 2007). The livestock sector was modeled in more detail based on livestock and region-specific feed baskets (Bodirsky et al., 2012; Schmitz et al., 2012; Weindl et al., 2010, 2015). Moreover, the scope of the model was further broadened by accounting for climate impacts on cropland and pasture productivity, their implications for land-use dynamics and agricultural production costs and possible adaptation options (Weindl et al., 2015). In addition, MAgPIE was extended by a comprehensive representation of

biomass and nitrogen flows in agriculture and upstream in the food supply chain, covering for example nitrogen budgets of cropland soils, the production and different uses of crop residues and conversion byproducts, animal waste management systems, and soil organic carbon accounting (Bodirsky et al., 2014, 2012). Moreover, while MAgPIE 1 only simulated a single baseline scenario, MAgPIE 2 translated the SRES storylines (Nakicenovic et al., 2000) into multiple scenarios with diverging drivers and scenario assumptions (Bodirsky et al., 2015, 2012). The representation of agricultural water use and water scarcity was strengthened by accounting for changes in irrigation efficiency over time (Schmitz et al., 2013) and by differentiating between green and blue water consumption (Biewald et al., 2014).

Structurally, the next evolution came with **version 3** introducing the concept of modules, allowing to split the code into thematic components and to have different realizations of the same component. Content-related extensions in version 3 were the introduction of afforestation as a climate mitigation measure that is endogenously calculated and incentivized by a tax on GHG emissions (Humpenöder et al., 2015, 2014), the endogenous simulation of future pasture area driven by feed demand and opportunity costs of grazing land (Popp et al., 2014), and dynamic feed baskets where feed efficiency and feed composition depend on livestock productivity trajectories (Weindl et al., 2017a, b). Model capacities with regard to agricultural water use were further improved by the inclusion of annual costs for irrigation (e.g. for water, fuel, labor and the maintenance of irrigation infrastructure), the exogenous representation of non-agricultural water demand for domestic use, industry and electricity production, the implementation of environmental flow requirements and the calculation of the annual volume of available irrigation water considering seasonal variations, growing periods of crops and water storage facilities provided by dams (Bonsch et al., 2014, 2015). The evaluation of climate impacts and mitigation measures was deepened across a broad range of studies using MAgPIE version 3, where an increasing emphasis was placed on socio-economic indicators such as food prices (Kreidenweis et al., 2016) and agricultural welfare (Stevanović et al., 2016). In addition, governance scenarios were incorporated into the model by using lending interest rates as discount rates to represent risk-accounting factors (Wang et al., 2016). The increasing complexity and scope of the model also allowed for multi-criteria sustainability assessments, e.g. regarding large-scale bioenergy production (Humpenöder et al., 2018). This is an important model feature that allows to address research questions in the context of the SDGs. The model was also used in the assessment of climate policy entry points to mitigation pathways consistent with the Paris Climate Agreement goals (UNFCCC, 2015). To that end MAgPIE was broadened to represent near-term policies given by nationally determined contributions (NDCs), and covering land-based national targets for avoiding deforestation and targeted afforestation (Kriegler et al., 2018).

Linked to the global gridded crop model LPJmL (Bondeau et al., 2007) and coupled with the energy and macro-economy model REMIND (Popp et al., 2011a), MAgPIE began to form the Potsdam Integrated Assessment Modeling (PIAM) framework (Kriegler and Lucht, 2015). MAgPIE 3 coupled with REMIND was among the Integrated Assessment Models (IAMs) that were applied to translate the story-lines of the SSPs into quantitative scenarios of possible societal developments, e.g. land-use and energy futures (Bauer et al., 2017; Kriegler et al., 2017; Popp et al., 2017).

## 2.2 New features in MAgPIE4

While the modularization concept was introduced with version 3, the code was only partly modularized and a full modularization was only achieved with **version 4** of the model. In addition to the modularization, version 4 increases spatial flexibility by introducing the concept of flexible regions. In addition to the flexible number of clusters within a world region it allows the user to freely choose the number and shape of world regions to be simulated in the model. While all previous model versions were limited to the regional aggregation introduced in version 1, it is now possible to choose a regional aggregation, with the country level (ISO 3166-1:2013) as the highest possible level of detail. The combination of full modularization and additional spatial flexibility in version 4 also marks the transition from model to modeling framework.

Content-wise MAgPIE 4 includes a new food-demand module, which couples MAgPIE 4 iteratively with a standalone food-demand model. The module estimates the distribution of body mass index, height and food intake by age-group, sex and country. Moreover, it estimates food waste and a more detailed dietary composition. For a given level of income, changes in food prices affect food demand through their effects on purchasing power. Furthermore, version 4 includes a more detailed representation of food processing.

Finally, version 4 is the first open source version of MAgPIE (Dietrich et al., 2018e). For this step proprietary data had to be separated from the model and code had to be cleaned and properly documented. All model dependencies which are required to run the model have also been published open source (gdx, magclass, madrat, mip, lucode, magpie4, magpiesets, lusweave, luscale, goxygen: Dietrich et al., 2018f, b, a; Klein et al., 2018; Dietrich et al., 2018g; Bodirsky et al., 2018a, b; Bonsch et al., 2018; Dietrich et al., 2018c; Dietrich and Karstens, 2018).

## 2.3 Modules

The MAgPIE 4 framework consists of 38 modules which are listed and briefly described below (by name and order as they appear in the code). A detailed description of each module and their realizations is part of the model documentation (Dietrich et al., 2018d). While some modules come with several realizations that are regularly exchanged for simulation runs, others remained mostly unchanged over time.

**drivers** Provides model drivers like population and income that are being used by multiple other modules.

**land** Simulates spatial competition of different land cover types for physical area.

**costs** Calculates total costs by summing up all costs in the model including production costs, investments into research and development or land expansion, tax expenditures, and mitigation costs.

**interest rate** Defines the interest rate based on the governance performance of a scenario storyline. The interest rate affects investment decisions in other modules (Wang et al., 2016).

**tc** Links investment into technological change to corresponding yield increases (Dietrich et al., 2014).

**yields** Estimates crop and pasture yields based on biophysical yield patterns from LPJmL and endogenous yield-increasing technological change (Dietrich et al., 2014). Biophyiscal patterns can optionally include climate change impacts (Stevanović et al., 2016).

**food** Estimates food demand and dietary composition on the country level based on population growth and economic development. The demand projections account for changes in the demographic structure (age, sex), physical activity, body mass index, body height, and food wasting patterns. Optionally, changed prices of agricultural commodities can reduce real income of consumers, resulting in elastic food demand.

**demand** Aggregates domestic demand for food, feed, seed, material and bioenergy usage as well as supply chain losses.

**production** Merges production values including crop-based production and livestock-based production into one production variable. Aggregates cellular production to the regional level for modules only interested in regional production levels.

**residues** Estimates crop residue biomass, recycling, burning as well as removal for feed and material usage. Estimates costs of residue removal (Bodirsky et al., 2012).

**processing** Simulates the processing of primary agricultural products into secondary products like sugar, oilcakes or ethanol, including processing costs.

**trade** Simulates trade between world regions based on cost competitiveness and historical trade patterns (Schmitz et al., 2012).

**crop** Simulates crop production and competition of different crop types for cropland, accounting also for crop rotation requirements. Estimates the terrestrial carbon pools of croplands.

**past** Estimates land dynamics and terrestrial carbon pools of pastures and range-lands.

**forestry** Simulates managed forests, including age-class dynamics, afforestation, and terrestrial carbon dynamics (Humpenöder et al., 2015, 2014).

**urban** Estimates dynamics of urban areas.

**natveg** Estimates dynamics of areas with natural vegetation, including natural forests.

**factor costs** Estimates the factor costs of crop cultivation, e.g. including costs for labor, machinery, or fuel. Costs for land, water, seeds, fertilizer, land conversion, and pollution certificates are accounted in other modules (Dietrich et al., 2014).

**landconversion** Calculates the costs for conversion between different land cover types.

**transport** Estimates intra-regional transport costs between farm-gate and proximate market center.

**area equipped for irrigation** Simulates the expansion of area equipped for irrigation and the related investment costs (Bonsch et al.,
25 2014).

**water demand** Estimates the demand for blue water to irrigate crops. Climate change impacts can be considered optionally (Bonsch et al., 2014).

**water availability** Estimates water availability for irrigation, accounting for natural runoff but also competing anthropogenic water usage. Climate change impacts can be considered optionally (Bonsch et al., 2015).

**climate** Provides information on climate zones for other modules.

**nr soil budget** Estimates cropland and pasture soil nitrogen budgets, including withdrawal of nutrients by harvested biomass, biological fixation, crop residues management, manure application, inorganic fertilizer, atmospheric deposition and soil organic matter loss (Bodirsky et al., 2012).

**nitrogen** Estimates nitrogen-related emissions in the forms of $N_2O$, $NH_3$, $NO_x$, $NO_3$-, and $N_2$ from managed soils and animal waste management (Bodirsky et al., 2012).

**carbon** Estimates terrestrial carbon stock changes and emissions, aggregating over different land cover types (Popp et al., 2014).

**methane** Estimates methane emissions from enteric fermentation, rice cultivation, and animal waste management.

**awms** Calculates the nutrient flows within animal waste management systems (awms) (Bodirsky et al., 2012).

**ghg policy** Simulates the impacts of taxing GHG emissions, air pollutants, and water pollutants. Estimates anticipated future benefits of mitigation (Humpenöder et al., 2014).

**maccs** Estimates the impact of GHG abatement technologies on emissions based on prescribed marginal abatement cost curves (maccs) and computes mitigation costs.

**som** Estimates the change in soil organic matter under changing land cover and soil management (Bodirsky et al., 2012).

**bioenergy** Derives the demand for 1st and 2nd generation bioenergy (Lotze-Campen et al., 2010; Klein et al., 2014).

**material** Derives the demand for non-energy material usage of bio-based products.

**livestock** Estimates the feed demand under consideration of the produced livestock products accounting for changing feed mix and feed conversion efficiencies under exogenous increases in livestock productivity. Estimates costs of livestock production, but excluding
costs for feed which are already accounted in other modules (Weindl et al., 2017a, b).

**disagg lvst** Distributes regional livestock production spatially among all cells belonging to this region by linking it to fodder or pasture production as well as urban areas.

**optimization** Minimizes total costs of the optimization problem for each time step using different optimization strategies to reduce run time.

Figure 1 provides a simplified visualization of the module interactions in the MAgPIE 4 framework. Simplification was required due to the vast number of existing interfaces and modules. Therefore, the figure only shows the most important linkages and modules or module groups in terms of relevance to the framework or representation of the underlying concept. An exact representation of all interfaces and modules can be found in the technical model documentation (Dietrich et al., 2018d). If modules are not directly linked it does not mean that they do not interact with each other. In some cases the feedback loops
go through a combination of modules rather than being direct links. An example is the livestock module which is triggering feed demand in the demand module, which is, via trade and production module, triggering production in the crop module.

## 3   Framework architecture

The framework consists of two layers. An outer layer written in R (R Core Team, 2017) handles the pre- and post-processing of data, manages and applies model configurations and initial calibrations. It also adjusts spatial resolutions of model runs
and organizes the parallel execution of run ensembles. It includes software libraries for code manipulation and analysis used for preparation and inspection of code in the inner layer (lucode: Dietrich et al., 2018g), packages for general data handling

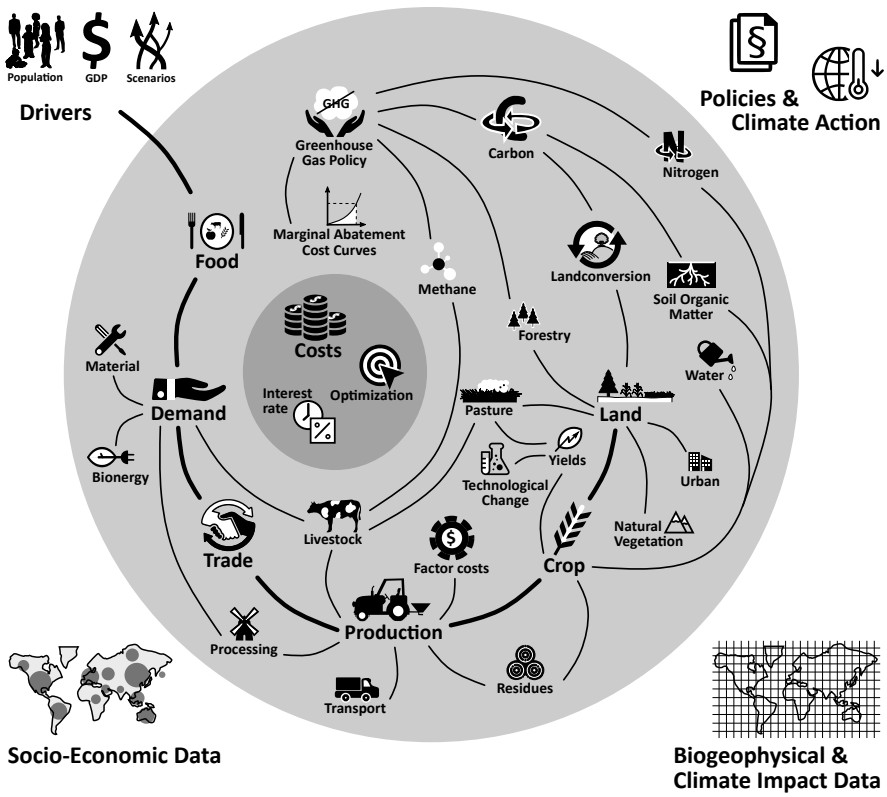

**Figure 1.** MAgPIE 4 framework with simplified modular structure and module interactions. See the model documentation (Dietrich et al., 2018d) for a more detailed presentation of module interactions and their implementations.

(magclass, lucode, madrat: Dietrich et al., 2018b, c, a), data analysis (gdx, magpie4, magpiesets: Dietrich et al., 2018f; Bodirsky et al., 2018a, b), documentation (goxygen: Dietrich and Karstens, 2018) and visualization (mip, lusweave: Klein et al., 2018; Bonsch et al., 2018). External packages provide tools for interfacing GAMS-specific output files (gdxrrw: Dirkse et al., 2016), data transfers (curl: Ooms, 2017) and extended visualization (ggplot2: Wickham, 2009). Most of the functions used in the
5   outer layer are not specifically bound to MAgPIE. They can also be used standalone and are therefore released as separate R packages.

The outer layer makes sure that model simulations can run in parallel and are portable and easily reproducible. Collections of runs can be written as R scripts with consecutive run execution statements. In each run execution a run composition process will apply the provided model configuration, create a run output folder and copy all relevant files to that folder.
10   The inner layer written in GAMS (GAMS Development Corporation, 2016) contains the optimization model with all its equations and constraints, the recursive dynamic logic which triggers the optimization for each time step consecutively and forwards results to the next time step and the code modularity implementation. The latter is assisted by the outer layer which is

monitoring code compliance and providing convenience functions for easier code manipulation in compliance with the modular structure (lucode: Dietrich et al., 2018g).

## 3.1 Modularity

Modularizing a model means separating the modeled system into multiple subsystems that exchange information only through clearly defined interfaces. Modularization helps to better comprehend the complex model and makes it easier to exchange or debug its components. Rather than having to think of the model as a single entity, it allows for separate conceptualizations of inter- and intra-module interactions.

The purpose and interface of each module is defined via a module contract. Model developers can expect that the module behaves according to the contract and design their implementations correspondingly. Developers of a module can design a realization with the contract solely as guideline, ignoring the rest of the model. Modularization disentangles model development and offers a safe method for model modifications under limited knowledge of the complete model.

Modularization allows for different representations of the same module, which we call realizations. For each model run, the model configuration defines which realization is activated for each module. Different realizations can vary in their representation of processes, assumptions, or level of detail, but not in their interfaces and general purpose defined in the module contract. Modularization therefore has the benefit to allow for module-comparisons. Different representations of a subsystem can be compared under ceteris paribus conditions for the rest of the model. This is a strong add-on to the current practice of model-comparison studies between different IAMs, where differences in subsystem dynamics can not be isolated due to differences in the overarching frameworks.

A module in MAgPIE is represented as a folder with realizations of the same module as sub-folders. Each sub-folder contains code and data required for its execution. Important for a modular structure is the existence of local environments. GAMS contains a single, global environment that allows each variable or parameter to be accessed from anywhere in the code. To emulate local environments a dedicated naming convention distinguishing local from global objects through a given prefix is employed. Code violations are avoided via support functions (Dietrich et al., 2018g) monitoring the code. Appendix A describes the technical detail of the modular implementation.

## 3.2 Reduced model feature

MAgPIE 4 is designed and modularized in a way that modules of the model can be excluded completely or single modules can run standalone. This might be the case for testing a specific module under perfect control of the incoming variables and parameters, or it might be an application for which only certain components play a role. This reduced specification can be then used to develop a module in a toy model environment before it is used in the full model, saving time and resources during development.

Technically, a standalone reduced model form is created by writing a separate main GAMS execution script which includes only a part of the existing modules. Interfaces which are outputs from modules excluded from the reduced model have to be provided by the reduced model main script. For example, food demand could be estimated in a reduced form only considering

population and income growth, but omitting the price feedback from the production side and thereby most other modules (See Appendix A and A1 for more information).

## 3.3   Model run composition

To allow for parallel execution of model runs and to improve reproducibility MAgPIE performs a model run composition. Purpose of the composition is to isolate the current model run before execution. Isolation is achieved by creating a separate output folder for each run in which all relevant data is copied. The main component of each output folder is a single GAMS file containing the full GAMS model and all inputs. This file is created by replacing all include statements in the original GAMS model code with corresponding input files or code segments. In case of conditional inclusions (e.g. realization selection) only the active inclusion is considered (e.g. the chosen realization). This approach leads to a fully self-contained GAMS file which can be shared and runs standalone. All other files in the output folder are supplementary and either used for run post-processing or provide additional information about the run setup (e.g. the run configuration file). For archiving it is recommended to store the whole output folder as an image of the respective run.

## 3.4   Flexible spatial resolution

The framework currently has two built-in spatial levels, a coarse level of world regions and a finer one of spatial clusters characterized by similar local characteristics on sub-regional level. Both levels are flexible in resolution.

The world regions in the model have the ISO 3166-1:2013 country level standard as a basis and allow for any aggregation of these countries to regions including keeping a single country as region. The finer resolution has a 0.5 degree spatial grid as reference which can be aggregated to clusters based on similar properties (Dietrich et al., 2013). The model outcomes at the cluster level can be downscaled back to the 0.5 degree grid in a post-processing step (Bodirsky et al., 2018a).

Input data pre-processing at ISO country or 0.5 degree level currently happens outside of the framework. An open source release will follow in that regard.

## 3.5   Documentation

Model documentation is based on the in-house developed toolkit goxygen (Dietrich and Karstens, 2018). Following the idea of the source code documentation generator tool doxygen (Heesch, 2008) it allows for documentation of the model via annotations in the model code itself. By extracting information from the code directly, such as variable declarations, equations definition or code snippets, it reduces the effort of writing the documentation and improves consistency between model documentation and code. By merging code and documentation text into one document the likelihood of out-of-sync code and documentation is reduced. The final MAgPIE 4 model documentation can be found online (Dietrich et al., 2018d).

## 3.6 Model evaluation

Model evaluation is performed with a validation database containing historical data and projections for most outputs returned by the model. After each model run, a validation report is generated automatically as PDF file. This report includes evaluation plots showing model outputs, historical data and other projections jointly for each output variable.

The automatically generated model evaluation documents for single model runs currently allow comparison of about 1,000 output variables with reference data. Comparison between model runs, i.e. between different scenarios, is rather difficult and inconvenient if the model results are scattered across different evaluation documents. To overcome this issue, we developed a) a routine for generating a single evaluation document with outputs for multiple model runs and b) the interactive scenario analysis and evaluation tools appMAgPIE and appMAgPIElocal (Dietrich and Humpenoeder, 2018), which show evaluation

plots for multiple scenarios including historical data and other projections based on an interactive selection of regions and variables. For illustration, we include selected evaluation plots in the results section and appendix (see Figures 2 and A1). The complete evaluation documents for all runs shown here are part of the supplemental material (Dietrich, 2019b).

## 4 Model Outputs

### 4.1 Impact of module realizations

Figure 2 shows three different applications of the flexible, modular structure in MAgPIE in comparison to a run with default settings. The first application (soil organic matter) is a case in which a model feature can be either switched on or off. While this module is slightly improving the overall accuracy of the model through improved fertilizer estimates it has high computational requirements, nearly doubling the run-time of the model. By default it is switched off but can be activated when needed, e.g. for studies focusing on fertilizer application. The second application (volume-based factor costs) is an example of a dispute about

the representation of a process, in this case the relationship between factor requirement costs and production. We compare here two realizations of factor requirement costs, one of which mainly links them to the area under production (default realization) and the other of which mainly links them to the production itself. As the available data sources did not allow to clearly link costs to area or production we were experimenting with different realizations of it. The flexible modular structure allowed to easily implement different hypotheses and compare them which each other. The third application (standalone food demand)

is an example in which a module is enabled to run standalone. Here, the food demand calculations, estimating regional food demand based on GDP projections and demographics, can also be run independent of other modules. This is especially useful for studies focusing on food demand itself or for general improvements in the projections itself.

    The evaluation plots show different stages and major components of a MAgPIE simulation. As figure 2 shows the population, which is an exogenous parameter driving the simulations, is identical for all four runs. As one of the drivers of food demand, the

population is also available in the food demand standalone case. We get a similar picture for the per capita food demand, which is the main output of the food demand model. The output is available for all runs and due to identical scenario assumptions also identical (for different assumptions see a variation across SSPs in Appendix A2).

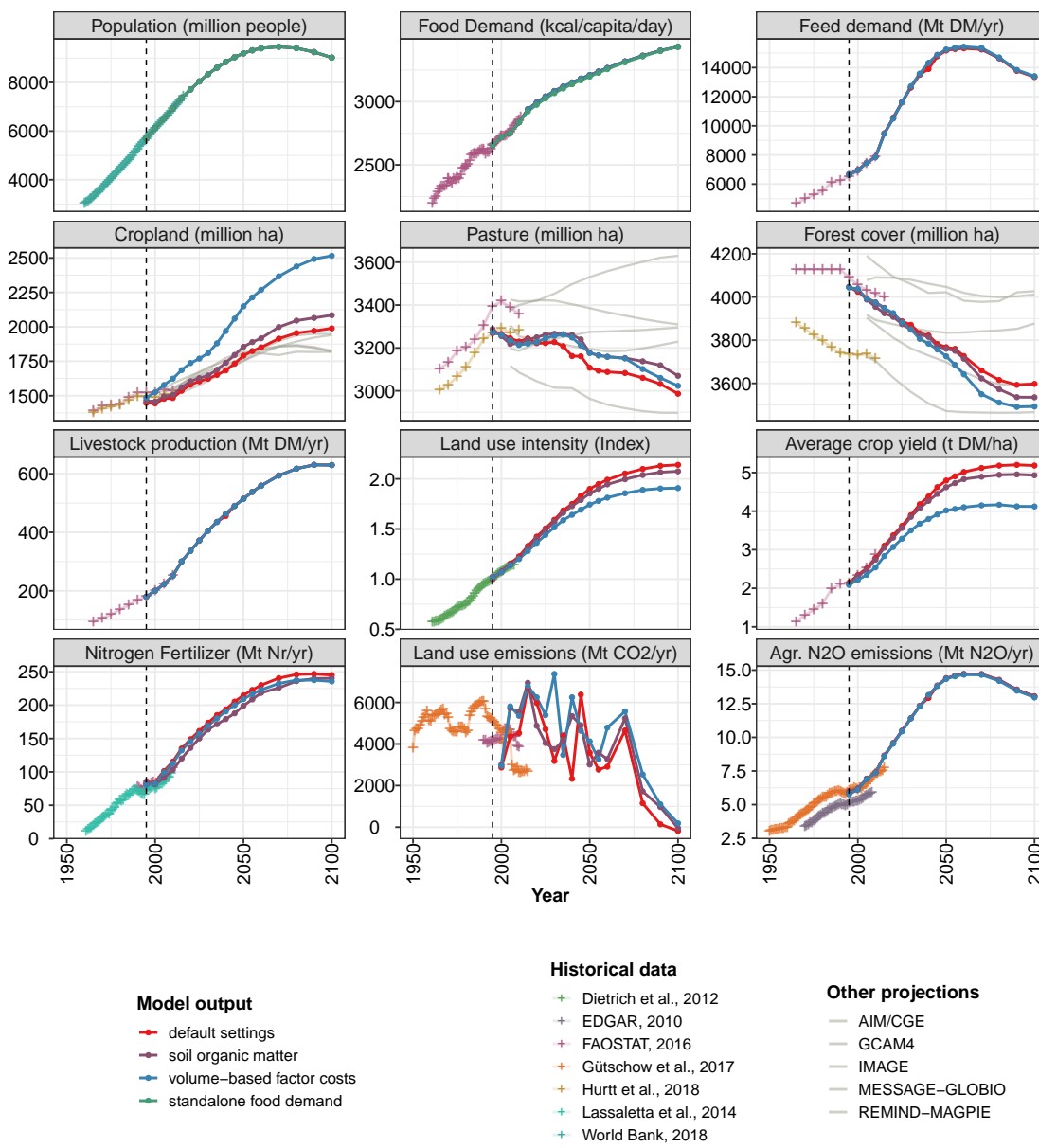

**Figure 2.** Evaluation plots for MAgPIE 4 inputs and outputs for the default settings, a run with soil organic matter explicitly modeled, a run with an alternative factor requirement setup with costs proportional to the production volume and a standalone run of the food demand module. Sources of historical data: (Dietrich et al., 2012; EDGAR, 2010; FAOSTAT, 2016; Gütschow et al., 2017; Hurtt et al., 2018; Lassaletta et al., 2014; World Bank, 2018). Sources of other projections for SSP2 reference scenario: (IAMC, 2016).

As all other aspects shown in the figure go beyond what is used or simulated in the food demand module, all remaining values could only be reported by the non-standalone runs. The combination of per capita food demand and total population provides the total food demand in the model which triggers total feed demand through consumption of livestock products. Also here the identical scenario assumption leads to the same results in all three runs. Differences can be observed in the global land cover and the productivity measures (land use intensity and average crop yields). Cropland shows higher expansion in the alternative scenarios compared to the default scenario while both scenarios show less intensification and lower yields. While the differences are rather small in the case of soil organic matter, the difference are quite pronounced in the alternative factor requirement case. In the case of soil organic matter this effect is triggered via the natural availability of nitrogen in the soil. Having SOM switched off the model assumes, that all required nitrogen is provided as fertilizer, while simulating SOM explicitly uncovers the already available nitrogen in the soil. This reduces the overall fertilizer requirements and slightly incentivizes land expansion as it gives the model access to more nitrogen. As the food demand is rather independent of this decision more land expansion leads to lower intensification requirements, lowering land use intensity as well as average yields. Having factor requirements primarily linked to the production rather than to the area on which it is produced strongly reduces the incentive in the model to intensify. Area dependent factor requirements strongly favor high yielding locations for production giving the model a strong incentive to concentrate production on high productive areas and to further boost productivity via intensification. Production dependent factor requirements on the other hand do not favor locations based on productivity making also rather unproductive areas interesting for production and thereby reducing the incentive for intensification. In combination this leads to significantly higher cropland expansion, higher forest reduction, less intensification and significantly lower crop yields. One can also observe that the difference in average yields is higher than in land use intensity, owing average yields to drop for two reasons: the lower land use intensification and the expansion into low productive areas.

CO2 emissions show strong fluctuations in all scenarios due to missing constraints linking carbon stocks with the goal function of the model (e.g. carbon pricing). This makes it in many cases an arbitrary decision for the optimizer to expand cropland into carbon rich or carbon poor areas. Besides its fluctuations the plot also shows higher overall emissions in the case of volume-based factor costs due to the overall higher expansion of cropland and reduction in forest areas.

## 4.2 Impact of spatial resolution

Figures 3 and 4 feature the spatial flexibility in MAgPIE 4. Compared are two scenarios with identical settings except for the spatial distribution of world regions and choice of clusters.

Figure 3 shows the default regional setup with 12 world regions[1] and 200 clusters. All regions are treated equally in the sense that the distribution of clusters among them follows the same rules and all regions are faced with the same type of constraints in the model.

Figure 4 shows a setup with a specific focus on Brazil. To gain higher spatial detail in Brazil it comes with a higher number of clusters in total. Brazil (BRA) is simulated as a world region together with its most important trade partners (Rest of Latin

---

[1]Canada, Australia & New Zealand: CAZ, China: CHA, European Union: EUR, India: IND, Japan: JPN, Latin America: LAM, Middle East and North Africa: MEA, Non-EU member states: NEU, Other Asia: OAS, Reforming countries: REF, Sub Saharan Africa: SSA, United States: USA

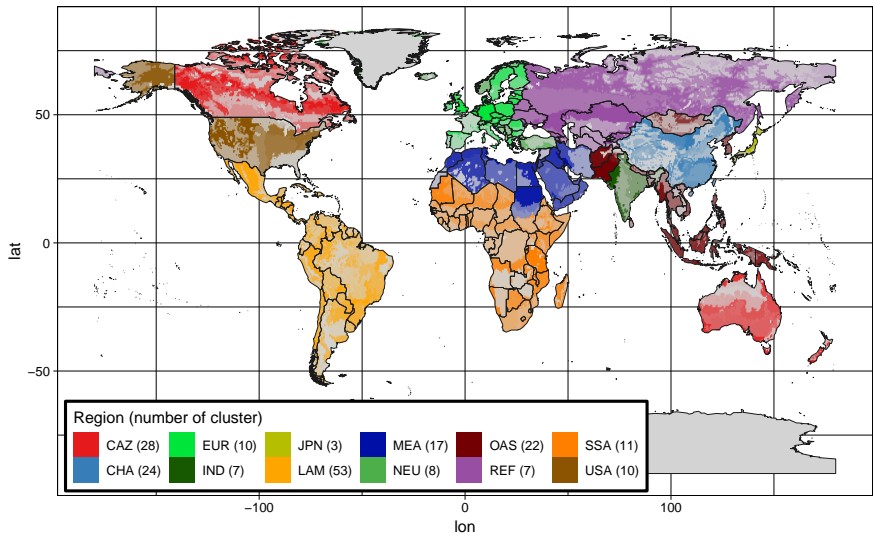

**Figure 3.** Standard MAgPIE 4 world regions and cluster setup: 12 equally treated world regions with 200 clusters in total

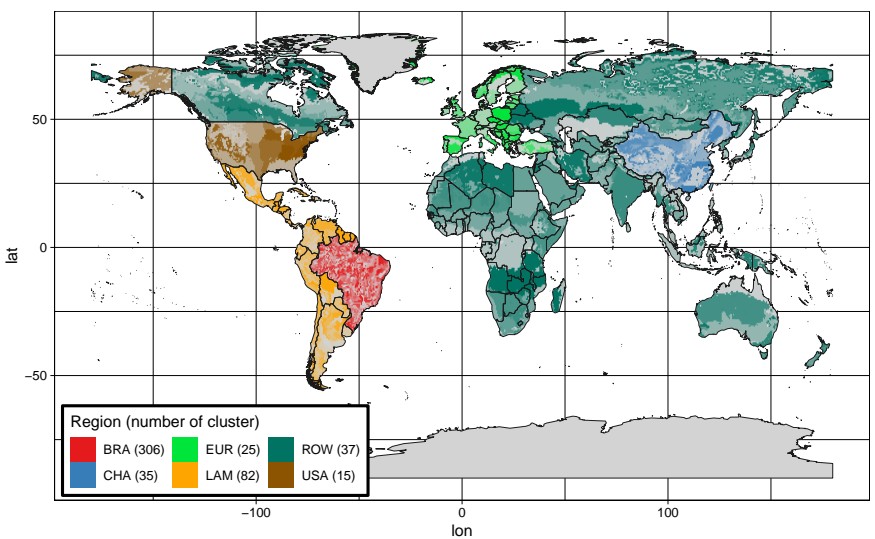

**Figure 4.** Study setup tailored to assessments with a focus on Brazil, with 6 world regions and 500 clusters : Brazil (BRA) in increased spatial resolution, its major trade partners Latin America (LAM), United States (USA), China (CHA) and Europe (EUR) in default resolution and rest of the world (ROW) combined to one region with reduced resolution.

America (LAM), United States (USA), China (CHA) and Europe (EUR)). Remaining countries, less relevant for a Brazil-centric study, are merged to a single region (ROW). Furthermore, the cluster allocation of 500 clusters in total has been shifted in favor of Brazil: Roughly four times more clusters are allocated to Brazil (306) compared to a default distribution of clusters.

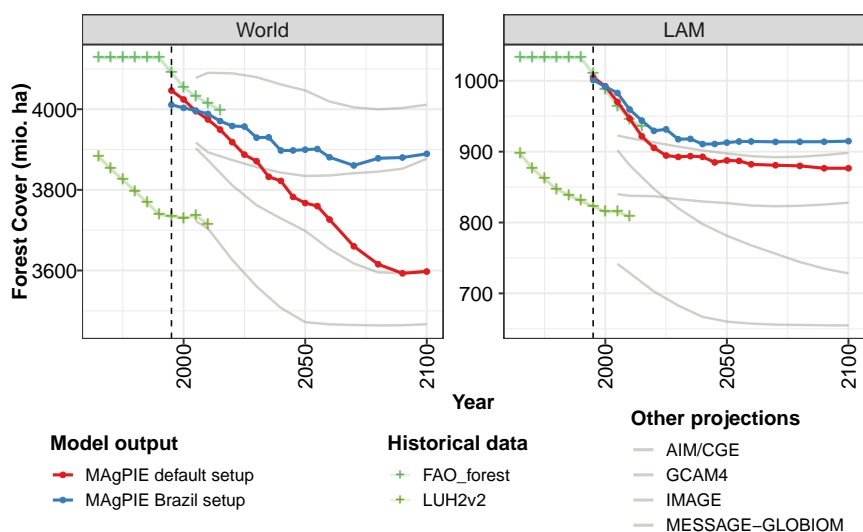

**Figure 5.** Comparison of global and Latin American forest cover with historical data sets and projections of other models.

At the same time the ROW region receives only roughly 0.7 times the number of clusters it would usually get (37), leaving room for a balanced number of clusters for all other regions. Detail gained for Brazil is attained with reduced detail for the rest of the world to keep the model complexity manageable for the applied solver.

Figure 5 shows the development of forest cover globally as well as for Latin America as a whole for both model setups. The plots show that the mapping has an effect on the overall forest cover development, both globally and regionally.

Comparison with historical data sets as well as projections on forest cover show that the differences between mappings are rather small compared to the overall uncertainty in these numbers. Nevertheless, a deeper look into the simulations uncovers that the global numbers of the Brazil-centric setup are unreliable as the reduced deforestation rate compared to the default setup is a consequence of the applied mapping. As the ROW region basically acts as a huge free trade region it can fulfill strong demand pressure coming from Sub-Saharan Africa with production from elsewhere, while trade limitations in the default setup limit this exchange and trigger deforestation within Sub-Saharan Africa (Dietrich, 2019b, compare m4p_default_validation.pdf p1558 and m4p_brazil_validation.pdf p1465).

In the case of LAM both runs show a rather similar picture in the aggregated forest cover projections for the region and it is not possible to clearly reject one of them. This is particular important as the regional aggregates in LAM are in the scope of both mappings and therefore should be sound. When choosing between them, one has to decide whether spatial details in Brazil or global trade patterns are the more decisive factor for accurate estimates of regional forest cover in LAM.

Looking at forest change patterns in Brazil and neighboring countries between 2000 and 2050 it becomes easier to introduce a ranking between the setups (Figure 6). While both settings show a tendency towards spatial specialization, this effect is much more pronounced in the default setup. Here, deforestation is nearly exclusively concentrated in Bolivia, Paraguay and South Brazil, along with strong reforestation in the Matopiba region (which in reality is Brazil's deforestation frontier), and without

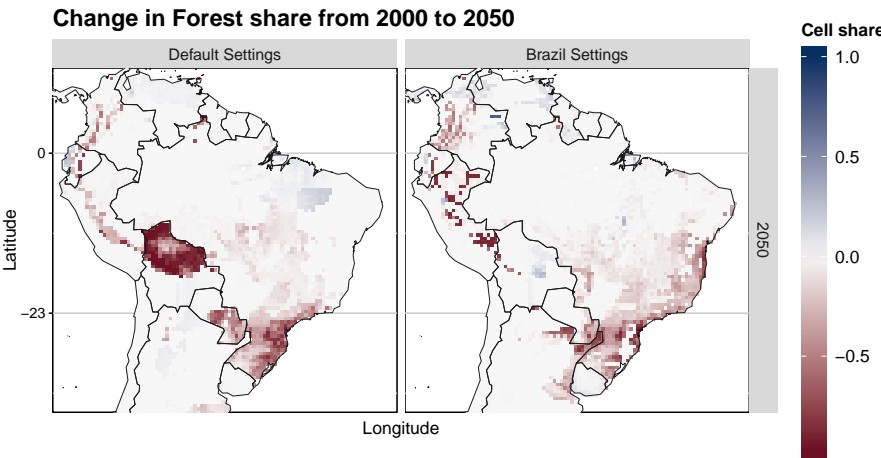

**Figure 6.** Comparison of changes in forest share from 2000 to 2050 in Brazil between default setup and Brazil setup.

deforestation in eastern Brazil. With Brazil-specific settings, the model shows a more balanced behavior. The big deforestation cluster in Bolivia disappears and while deforestation in Brazil primarily takes place in the South, it is less condensed and extends more to the North, which is more consistent with observations.

The observed specialization is a consequence of the homogeneous biophysical characteristics within each cluster which lead
5  to either-or-decisions in the model. It will either fully take a cluster into production or ignore it completely. In the default setup this effect is very pronounced due to the low number of clusters within Latin America. With more clusters, as in the Brazil setup, clusters better grasp the real spatial distributions of biophysical characteristics in the region and therefore lead to a more diverse picture. Whereas this effect is especially relevant for regional studies with focus on spatial patterns, it is less critical for global dynamics as long as the spatial aggregation is not introducing any systematic biases to the model.
10  While the Brazil setup improves the spatial representation of Brazil, it is only a first step as deforestation patterns show. As a second step towards a regional study, which is missing in this paper, it is always required to adopt regional distinctiveness into the model, such as region-specific policies relevant at this level of detail for this specific region.

## 5  Discussion & Conclusions

Since the first version of MAgPIE, the model has evolved from a crop-focused land-use allocation model to a modular open
15  source framework with a broad range of covered processes.

One main improvement introduced in MAgPIE 4 is the full code modularization. It is used as a tool to make the model better manageable as it structures the code in self-containing components which are interacting via interfaces with each other. It makes existing and missing interactions in the model better visible and allows to easily replace components by alternative

implementations. While the modular structure is rather intuitive for a system with loosely linked components one could argue that it might prevent a proper implementation of strongly integrated systems. Our experience is that, while the modular concept is working best for clearly separable systems, it also works in all other cases. The difference with strongly integrated systems is that the amount of interfaces and the required effort for developing new realizations are higher. Nevertheless, it still improves transparency in terms of model interactions and does not exclude any systems or dynamics from being represented in the model. Modules are also not static and the modular structure itself can and will also be changed if required. Modules might get created, deleted, merged or split over time. Module interfaces might get extended, reduced or modified. As both happens less frequently than changes within modules the modular structure can be best described as semi-static.

Besides modularization MAgPIE 4 introduces a series of other features such as automatic documentation of the GAMS code, the possibility to run parts of the model standalone, flexible spatial resolution and automatized creation of evaluation reports. The evaluation of selected model outputs shows that MAgPIE 4 projections connect well to historical data and projections from other modeling teams. Therefore, we consider MAgPIE 4 as an appropriate tool for simulating scenarios of future land-use. The case study with higher spatial resolution for Brazil demonstrates how the flexible spatial resolution approach works and how it can be meaningfully applied for research questions with a regional focus. With the open-source publication of the MAgPIE 4 model code, we aim to increase the transparency and reproducibility of model experiments for reviewers, stakeholders and other interested groups. Furthermore, we expect that the future development of the MAgPIE modeling framework will benefit from cooperation with individuals and other research institutions, as enabled by the open-source availability of the code.

*Code and data availability.* The MAgPIE code is available under the GNU Affero General Public License, version 3, (AGPLv3) via GitHub (https://github.com/magpiemodel/magpie). The release version 4 used in this paper can be found via Zenodo (https://dx.doi.org/10.5281/zenodo.1445533). The technical model documentation is available under https://rse.pik-potsdam.de/doc/magpie/version4/ and also archived via Zenodo (https://doi.org/10.5281/zenodo.1471526). Test runs shown in this paper are archived at http://dx.doi.org/10.5281/zenodo.2572620 and corresponding evaluation documents can be found at http://dx.doi.org/10.5281/zenodo.2572581.

## Appendix A: Modular GAMS code

The aim of a modular GAMS code is to separate different parts of the model code from each other and to set the interaction rules between each other. Usually, such a separation is achieved via local environments. If information should be transferred from one module to another this has to be done explicitly via a global environment which is visible to all modules. The global environment acts as an interface between modules. GAMS does not distinguish between environments. All objects are accessible from everywhere in the code. To emulate local environments we introduced a naming convention indicating whether the object should be treated as global or local. Each object is required to have a prefix in its name indicating what type of object it is (e.g. "v" for variable or "p" for parameter) and to which environment it belongs (local or global). While elements in the global environment are marked with an "m" (module interface), elements in local environments carry a number in its prefix that is unique for every module. In this naming convention "vm_area" represents for instance a global (m) variable (v)

**Table A1.** Module Components

| component | description |
| --- | --- |
| sets | set declarations |
| declarations | variable, equation and parameter declarations |
| inputs | read-in of file inputs |
| equations | equation definitions |
| preloop | calculation before time step loop starts |
| presolve | calculation in the time step loop before the solve statement |
| postsolve | calculation in the time step loop after the solve statement |

containing area information, while "p42_costs" is a local parameter (p) of module 42 containing cost information. While local objects are technically still accessible from everywhere in the code, they are formally only allowed to be accessed from within the corresponding module. In MAgPIE 4 the proper use of the naming convention is ensured by the R function codeCheck in package lucode (Dietrich et al., 2018g). The function runs at the beginning of every model simulation and either warns or even

stops the run in case of code violations.

Each module in MAgPIE comes with a module contract that can be found at the beginning of the documentation for each module. The contract consists of three components: *task description*, *required inputs* and *promised outputs*.

The *task description* defines the purpose of the module. The *list of inputs* defines which inputs the module expects in order to be able to perform its tasks. The *output list* defines the information the module will provide to the rest of the model. The

contract contains all information that is necessary to be able to work with the module or to develop it. It therefore reduces the need to understand the model as a whole. The contract approach is similar to the function concept in other languages. The difference in GAMS is that a module cannot be run at once but is split up into topic-wise chunks and distributed over the whole model run. Table A1 lists the most relevant module chunks in MAgPIE.

In the first chunk, each module can introduce its own sets. Similarly, the declarations of parameters, variables and equations

of all modules follow as a second chunk. All other chunks follow with the same principle. This split-up into chunks allows modules to interact at different stages of the run. They can, for instance, exchange information before the model is solved and exchange another set of information after the model has been solved. Technically, this is implemented via an include file, which is going through all modules for each chunk checking whether a module provides a code piece to the given chunk and if so includes it.

The modular concept also allows to introduce alternative versions of a module, called "realization". Similarly to the include file, each module comes with a GAMS file including a realization based on the choice in the configuration of the model. Different realizations are implemented as alternative folders in the corresponding module. The implementation of a realization

is only bound by the module contract. This implies that it must be able to perform its calculations based on the promised inputs and must provide the promised outputs. This level of freedom allows to have very different realizations of a module.

## A1    Reduced model feature

When developing a module realization, it might be handy not to have to run a full-feature model simulation, but rather a reduced version of the model. To slightly reduce model complexity, all modules can be switched to their simplest realization and the spatial resolution of the model can be reduced. If the rest of the model should rather be reflected as a toy model with very limited complexity, the reduced model feature can be used. As each module defines which inputs it needs for the run via its module contract, it is also possible to write a dummy model that only provides these inputs to the module and handles the outputs it receives from the module that should be run standalone. This can be handy if a module is to be tested under well-defined boundary conditions or if a study purely focuses on a sub-component of the model.

In MAgPIE, such a reduced model version is created by adding a corresponding dummy model to the "standalone" folder of the model. The dummy model includes the module that should run standalone and ensures that all interfaces of the module are properly addressed. The reduced model itself can be run again via the Standard R interface. Only the name of the model (cfg$model) has to be changed in the configuration file from main.gms to the name of the new dummy model.

## A2    Key evaluation examples

For the model evaluation, we set up an extensive database with historical and projected data for the various outputs the model can produce. In Figure A1, we show evaluation plots for 12 model inputs and outputs and 5 Shared Socioeconomic Pathway (SSP) scenarios at global level (O'Neill et al., 2017). A complete evaluation output for all scenarios shown in this paper can be found in the supplementary material (Dietrich, 2019b). The purpose of Figure A1 is threefold: First, the figure illustrates how the evaluation plot is structured. Second, the evaluation plot for each key land-use variable demonstrates the model performance compared to historical data and other projections at global level. Third, the figure shows how contrasting scenario assumptions based on SSP 1-5 shape model outputs.

Note that the first three evaluation plots in Figure A1, population, food demand and feed demand, show model drivers, while the other nine evaluation plots show endogenous model outputs. Checking consistency of the model drivers is done via comparison to alternative data sources. For instance, population projections are taken from the SSP database. Comparing these projections to historical population data from the World Bank (World Bank, 2018) shows that both data sets match with respect to levels and trends for the period 1995-2015. While population is a completely exogenous driver, food and feed demand are calculated endogenously in the model, but calibrated to FAOSTAT (FAOSTAT, 2016) until the year 2010. Here, the evaluation plots for food and feed demand show that the calibration routine works as expected and that projections for the coming decades continue recent trends.

Spatially explicit land cover in MAgPIE 4 is initialized with a modified version of the LUH2v2 data-set for the year 2000 (Hurtt et al., 2018). The main modification is calibration of forest cover to data provided by FAOSTAT at country level. Overall, the land cover dynamics for cropland, pasture and forest produced by the model framework for the period 1995-

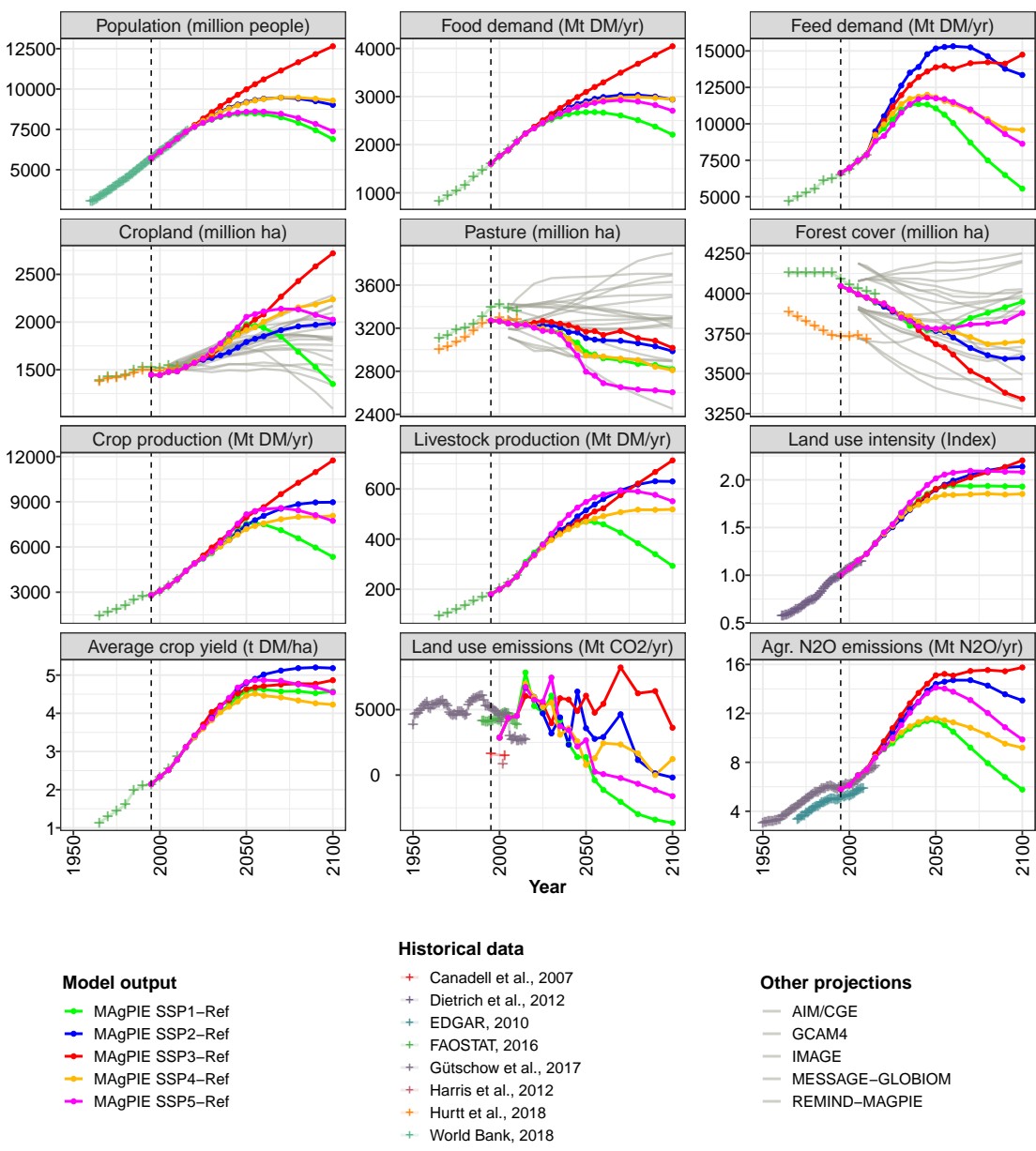

**Figure A1.** Evaluation plots for MAgPIE 4 inputs and outputs for SSP1-5 reference scenarios at global level. Sources of historical data: (Canadell et al., 2007; Dietrich et al., 2012; EDGAR, 2010; FAOSTAT, 2016; Harris et al., 2012; Hurtt et al., 2018; Gütschow et al., 2017; World Bank, 2018). Sources of other projections for SSP1-5 reference scenarios: (IAMC, 2016).

2015 are comparable with respect to level and trend to LUH2v2 and FAOSTAT (Figure A1). The land cover projections until 2100 for the five SSP reference scenarios (SSP1-5) mainly depend on the underlying socio-economic assumptions because

these reference scenarios include only currently implemented climate policies, but not ambitious climate polices such as the global carbon prices needed for the 1.5 or 2 degree target. For instance, the SSP3 "regional rivalry" scenario with the strongest population growth and limited trade reflects highest cropland expansion and deforestation. In contrast, the SSP1 "sustainability" scenario with declining world population after 2050 and globalized trade shows a decline in cropland after 2050 along with regrowth of forests.

The evaluation plots for cropland, pasture and forest also show projections from other models for SSP1-5 reference scenarios (IAMC, 2016). With some exceptions (e.g. cropland expansion in SSP3), the MAgPIE 4 projections for cropland, pasture and forest are mostly within the range of these other projections. Land-use intensity and average crop yields projected by MAgPIE 4 compare well to historical data with respect to level and trend. Annual $CO_2$ emissions from land-use change is a highly uncertain variable, which is illustrated by the spread of the four different historical sources (Canadell et al., 2007; FAOSTAT, 2016; Harris et al., 2012; Gütschow et al., 2017) included in the respective evaluation plot (Figure A1). The MAgPIE 4 projections for annual land-use change emissions start at the upper end of these historical data, and develop in the future in line with the projected land cover dynamics. For instance, land-use change emissions in the SSP3 scenario remain rather constant until 2100 due to ongoing deforestation for cropland expansion. In contrast, $CO_2$ emissions in the SSP2 "middle of the road" scenario decline towards zero by 2100, and even become negative in SSP1 after 2050 due to regrowth of forests. Finally, the agricultural $N_2O$ emissions show again good agreement in level and current trend with comparison data. While projections in SSP1 and SSP4 show a continuation in trend till 2050, all other SSP projections show a steeper increase in emissions in this time frame compared to historical observations. All projections have in common that they project a significant change in trend around 2050 with declining emissions in all scenarios except of SSP3 in which emissions continue to increase but at a lower speed.

More information information about the runs can be found in the corresponding evaluation documents (Dietrich, 2019b) and model runs (Dietrich, 2019a). The latter contains for instance NetCDF-files with spatial land cover information of the corresponding runs (cell.land_0.5.nc).

*Author contributions.* HLC wrote the original model. AP and HLC guided the model development. JPD developed and implemented the framework structure (modularity, spatial flexibility, code-based documentation). JPD, BLB, IW, FH, MS, KK, UK, XW, AM, DK, GA, AWY, EA, LB, SW and AG prepared input data. JPD, BLB, IW, FH, MS, KK, UK, XW, AM, DK, GA, AWY, EA and HLC developed the content of the model framework. JPD, AG, DK and LB provided technical support for the development. JPD, BLB, IW, FH, MS, KK, XW, AM, DK, GA, AWY, EA, FB, DC and AP wrote the model documentation. JPD managed the Open Source release. JPD, IW, MS, DK and FH wrote the manuscript. KK, AM, BLB and JPD developed the model schematic. JPD, BLB, FH, GA and EA designed the output examples. All authors prepared the model framework for release, discussed the manuscript and supported the writing of the article.

*Competing interests.* The authors declare that they have no conflict of interest.

*Acknowledgements.* The authors thank for the data provided by FAOSTAT, Worldbank, and the SSP scenario modelers.

We thank Christoph Müller, Elmar Kriegler, Susanne Rolinski, Nico Bauer, Gunnar Luderer and colleagues at PIK for valuable discussions during the development of the modeling framework. We thank Joshua Elliot and Todd Munson for their support in improving the model optimization process in the framework.

5      The research leading to these results has received funding from the European Union's Horizon 2020 research and innovation programme under grant agreement No 689150 (SIM4NEXUS),No 776479 (COACCH) and No 652615 (SUSTAg via the FACCE SURPLUS framework FKZ 031B0170A). This work was also supported by ENavi (FKZ 03SFK4B1), one of the four Kopernikus Projects for the Energy Transition funded by the German Federal Ministry of Education and Research (BMBF). We acknowledge the doctoral scholarship for GA granted by Fundação de Amparo à Pesquisa do Estado de Minas Gerais (FAPEMIG), and the scholarship for EA from CAPES / Programa de Doutorado

10    Sanduíche no Exterior / Process number 88881.135263/2016-01. We also acknowledge Leibniz Association's Economic Growths Impacts of Climate Change (ENGAGE) project under grant number SAW-2016-PIK-1 which funded the research of AM. The work of KK was funded by the DFG Priority Program "Climate Engineering: Risks, Challenges, Opportunities?" (SPP 1689) and specifically the CEMICS2 project (grant number ED78/3-2).

Lastly, we thank the three anonymous reviewers for their valuable remarks which led to significant improvements of the paper.

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
