# Peer review of "MAgPIE 4 - A modular open source framework for modeling global land-systems"

_Geoscientific Model Development, 2018_

## Referee Comment (RC1) · Anonymous Referee #1 · 30 Dec 2018

The manuscript reflects an impressive effort: taking an existing model and turning it into a framework while meeting the demanding requirements of open sourcing it (licensing, distribution, documentation, and so on). The manuscript is a well-structured overview of the MAgPIE 4 framework.

In places, the manuscript can benefit from clarification and polish:

page1_line8: The abstract lists "flexible detail in process dynamics" as a feature. In the main text this phrasing does not recur, and it is unclear what it refers to: adjustable temporal resolution? Otherwise? Modify to bring the abstract in harmony with the content.

page2_lines29-30 "It also means that the complexity of a module realization can be

chosen based on the importance of this component for the given question". I presume this refers to the freedom of choosing between different realizations of a module, picking one with a degree of complexity sufficient for the task at hand. If so, the phrasing is incorrect since "the complexity of a module realization" is fixed and hence can not be chosen. Rephrase.

page3_lines6 The sentence starting with "An output" is confusing. Suggestion: The main text is completed by an output section – showing some select model output and a specific use case of the spatial flexibility provided by the framework – as well as a conclusions and outlook section.

page8_lines17-18 Imply that the modularity is implemented in GAMS: "The inner layer written in GAMS (...) including the code modularity implementation". As explained in appendix A, the modularity is in part enabled by a naming convention as GAMS lacks name spaces, and in part by R code to check that the naming convention is adhered to. This extends beyond what GAMS provides. Moreover, it is the reviewers understanding that further R functionality is used to compose the chosen module realizations written in GAMS to a single GAMS source file. As such, it is inaccurate to imply that the modularity is implemented in GAMS. Rather the modularity results from extending GAMS with a naming convention and R helper code. Please reflect this in the text. Some words on the composition would also be welcome: much emphasis is put on the modularity of the framework, so the text should reflect it accurately and completely.

page8_line21 "a physical separation of the respective model code". Presumably this is meant to reflect the organization of the model code in directories and files. If so, using the word physical here obfuscates the matter, and is not accurate given the many layers of indirection between logical and physical storage in modern computing systems. Suggestion: "a hierarchical organization of the respective model code"

page9_line3 "Physically a module..." Similar concern as above. Suggestion: "A module in MAgPIE is represented as a folder..."

page10_section3.5 discusses the model evaluation. Specifically, line 9 mentions "The automatized model evaluation documents currently validate". As written, this suggests that the documents are automatized and perform validation. From the preceding text, it is clear that instead the PDF evaluation documents are automatically generated, in principle allowing for human evaluation, though, at 2000 pages, practice is unlikely to reflect principle. Rephrase. Suggestion: "The automatically generated model evaluation documents currently allow comparison of about 1,000 output variables with reference data".

page13_line4- The paragraph discusses a revised setup emphasizing Brazil, but reducing the number of clusters elsewhere. It seems implied, but is not explicitly stated, that this serves to keep resource usage tractable or constant. This paragraph can benefit from clarification and more lucid phrasing.

page15_fig5/page13_lines11- Some words on the causative mechanisms for the marked outcome difference between the default and Brazil setup would be welcome.

Correction suggestions for spelling/syntax/punctuation:

page1_line5 computationally intensive page3_line27 region-specific page5_line1 choose a regional aggregation, with the country level () as the highest.... page5_lines4&5 food-demand page8_line28 realizations page9_line25 The model outcomes at the cluster level page9_line27 data pre-processing at ISO country or 0.5 degree level page13_line4 less -> fewer page16_line11 and other research institutions, as enabled by

---

## Referee Comment (RC2) · Anonymous Referee #2 · 3 Jan 2019

The paper presents the IT architecture of the MAgPIE framework focusing on two features: modularity and flexibility of spatial resolution. This is a rather technical paper which is well written and easily understandable in spite of its technicity. This type of paper is welcomed to improve the transparency of models and help interpreting their result.

Here are my comments:

1. The presentation of the modules (p. 5-7) raise a number of issues: (i) The definition of the modules is sometimes vague. The "costs" module is not easy to grasp: what kind of aggregates does it make? In fact, we wonder why this is a separate module for it. Why is the agregation not done in the corresponding module? The "production" module is defined as aggregating cellular production to the regional level, but how

the cellular production is defined? I would say this result from the "yields" and "crop" module, but this is not clear from the text and from Figure 1.

(ii) Prices are almost absent from the picture while they are a key element of the system. They are the primary drivers of the intensification mechanisms which are for this reason unclear here: is there some livestock intensification? Does the technological change react to price or it is exogenous? Also how is the fertilizer use treated? In so doing we don't see the substitution possibilities between production factor while this is basically what the model represent.

(iii) Finally some feedback loops seem to be lacking, e.g.: the production of residues should affect the bioenergy module; the crop module should affect the livestock throug feed production; the livestock production may affect the yield through manure and the availabilty of land may have an impact on yields.

The last two points reveal the difficulty of representing a system in a modular way, as each module strongly interacts with the other, making the frontier between them sometimes meaningless. Livestock and crop production system are typical examples as they are generally strongly integrated. This point is an important barrier to the modular representation which should be discuss in deepth and better justified. In some cases the modular representation is appropriate for modeling reality, however in some cases it could put into question the consistency of the model. Most importantly, this approach may be seen as only compatible with conventional agriculture, and not with alternative agricultural systems promoting a systemic approach.

2. The key evaluation examples are not very informative in the context of this paper. The paper presents the "framework" and not really the model (i.e., the economic and biophysical mechanisms represented). For this reason, having, e.g., an evaluation of the crop yield simulated by the model is not very relevant to the paper. We would rather expect an evaluation of the modular architecture, as the authors did for the spatial resolution in the next section. How the modules perform running together vs standalone

?

3. The evaluation of the spatial flexibility is interesting however I did not really understand why there is so much difference in the default and Brazil-specific settings. Why is there much specialization in the default setting? And to what extent does it affect the spatial deforestation/reforestation pattern in Brazil? Also, the paragraph beginning on p13 l4 is quite difficult to understand (why is there 200 clusters in the default version and 500 in the region-specific one ?)

---

## Referee Comment (RC3) · Anonymous Referee #3 · 7 Jan 2019

**General Comments**

This paper describes the model MAgPIE 4, its history and release as an open source modelling framework. It serves as an introduction to the modelling framework as it is released, and as such it is well-suited as a citation for future users. There is good explanation of the structure of the model, including its modularity and references to further documentation. However, the effects of running modules under different setups is not explored. The article also presents a case study with a focus on Brazil that demonstrates the regional flexibility of the framework and how different regional definitions may affect results. This last topic is particularly interesting and deserves to be expanded a bit. In particular what may be behind the different results that arise from changing regional definitions. Finally, the absence of any description of the objective

function of the model framework is an omission that should be corrected.

Therefore, I recommend that this article be accepted with minor revisions as detailed below.

Specific Comments

There is much emphasis on the modularity structure of the framework, but the article does not go beyond description of this modularity. It lacks any mention of how choices of modular setup may affect results. For example, how does a module behave in standalone versus integrated mode? Or how does changing a specific module realization affect other modules? This analysis should be included here as, for example, a case study of one specific module. Even if this is performed in a separate article or online documentation resource, a quick summary of the results of such an experiment should be included for illustrative purposes.

The optimization methodology should be better explained. There is only a brief mention of the method in the description of the optimization module, which states that the model "minimizes total system cost" (p7, l14). How is this done? Is the model dynamic recursive? This has been explained in other articles using MAgPIE, but it should be included here, either in the main text or the appendices. A description of the objective function and the optimization method is in order.

Another important issue is the expansion of the discussion on what drives the changes in results from different regional aggregation. In particular, the difference in global forest cover changes by about 10% when using the Brazil setup should be explored in more detail. Even if it is simply the result of coarser resolution in the ROW region, it would be interesting to hear more about the interpretation of these results. Is 10% an acceptable uncertainty level? Which global regions are most affected by the changing the regional definitions? Why? This would expand the discussion section as well.

Also, the initial land cover map most likely plays an important role on the future projections. As such, it would be advisable to include a figure with the base year land cover map, even though this may be extracted from Hurtt et al. (2018). In fact, land cover maps for other milestone years the authors deem important may also help the user to understand model dynamics.

Technical Comments

P13 l4: repeated "with"

---

## Author Comment (AC1) · 20 Feb 2019

We would like to thank the Referee for the time spent reviewing the paper and the given remarks which in our opinion were without exception useful and helped to significantly improve the paper.

**[REFEREE COMMENT 1]: The manuscript reflects an impressive effort: taking an existing model and turning it into a framework while meeting the demanding requirements of open sourcing it (licensing, distribution, documentation, and so on). The manuscript is a well-structured overview of the MAgPIE 4 framework. In places, the manuscript can benefit from clarification and polish: page1_line8: The abstract lists "flexible detail in process dynamics" as a feature. In the main**

[Figure]

**text this phrasing does not recur, and it is unclear what it refers to: adjustable temporal resolution? Otherwise? Modify to bring the abstract in harmony with the content.**

[AUTHORS RESPONSE 1]: "flexible detail in process dynamics" was meant to refer to exchangeability of different realizations of a module as this is not necessarily the case just because code is modular. We rephrased it to "modular structure with exchangeable module implementations" avoiding the term "realization" as its definition is coming later in the text.

**[REFEREE COMMENT 2]: page2_lines29-30 "It also means that the complexity of a module realization can be chosen based on the importance of this component for the given question". I presume this refers to the freedom of choosing between different realizations of a module, picking one with a degree of complexity sufficient for the task at hand. If so, the phrasing is incorrect since "the complexity of a module realization" is fixed and hence can not be chosen. Rephrase.**

[AUTHORS RESPONSE 2]: We rephrased the paragraph:

*[TEXT EDIT 2] "Flexibility in the level of detail means adjusting the temporal and spatial resolution. It also means that module realizations can be chosen based on the research question and thereby adjusting the model complexity appropriately."*

**[REFEREE COMMENT 3]: page3_lines6 The sentence starting with "An output" is confusing. Suggestion: The main text is completed by an output section – showing some select model output and a specific use case of the spatial flexibility provided by the framework – as well as a conclusions and outlook section.**

[AUTHORS RESPONSE 4]: We modified the text as suggested.

**[REFEREE COMMENT 5]: page8_lines17-18 Imply that the modularity is implemented in GAMS: "The inner layer written in GAMS (...) including the code modularity implementation". As explained in appendix A, the modularity is in part**

**enabled by a naming convention as GAMS lacks name spaces, and in part by R code to check that the naming convention is adhered to. This extends beyond what GAMS provides. Moreover, it is the reviewers understanding that further R functionality is used to compose the chosen module realizations written in GAMS to a single GAMS source file. As such, it is inaccurate to imply that the modularity is implemented in GAMS. Rather the modularity results from extending GAMS with a naming convention and R helper code. Please reflect this in the text. Some words on the composition would also be welcome: much emphasis is put on the modularity of the framework, so the text should reflect it accurately and completely.**

[AUTHORS RESPONSE 5]: Strictly speaking, the statement that the modularity implementation is part of the inner layer is not wrong. All rules developed to achieve the modular structure are applied to the GAMS code itself and can be handled without additional support from external functions. However, it is correct that in MAgPIE the outer layer is adding functionality to the modular structure, especially monitoring the proper application of modularization rules to the GAMS code. Without the outer layer, it would be rather complicated to detect code violations. Besides it we also have some convenience functions in R making it easier to manipulate the GAMS code in a compliant way to the given modularization rules. To make it more transparent we kept the sentence about the inner layer as it was, but added some information about the supporting extensions coming from the outer layer:

*[TEXT EDIT 5] "The inner layer written in GAMS contains the optimization model with all its equations and constraints, including the code modularity implementation. The latter is assisted by the outer layer which is monitoring code compliance and providing convenience functions for easier code manipulation in compliance with the modular structure (lucode)."*

[AUTHORS RESPONSE 5]: The run composition to a single GAMS file mentioned by the reviewer is another feature coming from the outer layer which is attached to the

modular structure. However, the modular structure also works properly if the runs are not composed to a single file. Hence, this feature is an independent feature focussing on parallelization and reproducibility of model runs. As suggested by the reviewer, we extended the description of the model run composition by adding a dedicated subsection "Model run composition" to the framework description:

*[TEXT EDIT 5] "To allow for parallel execution of model runs and to improve reproducibility MAgPIE performs a model run composition. Purpose of the composition is to isolate the current model run before execution. Isolation is achieved by creating a separate output folder for each run in which all relevant data is copied. The main component of each output folder is a single GAMS file containing the full GAMS model and all inputs. This file is created by replacing all include statements in the original GAMS model code with corresponding input files or code segments. In case of conditional inclusions (e.g. realization selection) only the active inclusion is considered (e.g. the chosen realization). This approach leads to a fully self-contained GAMS file which can be shared and runs standalone. All other files in the output folder are supplementary and either used for run post-processing or provide additional information about the run setup (e.g. the run configuration file). For archiving it is recommended to store the whole output folder as an image of the respective run."*

**[REFEREE COMMENT 6]: page8_line21 "a physical separation of the respective model code". Presumably this is meant to reflect the organization of the model code in directories and files. If so, using the word physical here obfuscates the matter, and is not accurate given the many layers of indirection between logical and physical storage in modern computing systems. Suggestion: "a hierarchical organization of the respective model code"**

[AUTHORS RESPONSE 6]: We removed the sentence as it was indeed misleading and not providing any additional insight.

**[REFEREE COMMENT 7]: page9_line3 "Physically a module..." Similar concern**

**as above. Suggestion: "A module in MAgPIE is represented as a folder..."**

[AUTHORS RESPONSE 7]: We modified the text as suggested.

**[REFEREE COMMENT 8]: page10_section3.5 discusses the model evaluation. Specifically, line 9 mentions "The automatized model evaluation documents currently validate". As written, this suggests that the documents are automatized and perform validation. From the preceding text, it is clear that instead the PDF evaluation documents are automatically generated, in principle allowing for human evaluation, though, at 2000 pages, practice is unlikely to reflect principle. Rephrase. Suggestion: "The automatically generated model evaluation documents currently allow comparison of about 1,000 output variables with reference data".**

[AUTHORS RESPONSE 8]: This is a valid objection. We corrected the description of evaluation documents and improved the wording in the whole paragraph.

**[REFEREE COMMENT 9]: page13_line4- The paragraph discusses a revised setup emphasizing Brazil, but reducing the number of clusters elsewhere. It seems implied, but is not explicitly stated, that this serves to keep resource usage tractable or constant. This paragraph can benefit from clarification and more lucid phrasing.**

[AUTHORS RESPONSE 9]: The assumption is correct. In our specific case the applied solver limits the complexity because with higher complexity it will be not able to compute a solution to the problem. We made this limitation now explicit in the text:

*[TEXT EDIT 9] "Detail gained for Brazil has to be bought with reduced detail for the rest of the world to keep the model complexity manageable for the applied solver."*

**[REFEREE COMMENT 10]: page15_fig5/page13_lines11- Some words on the causative mechanisms for the marked outcome difference between the default and Brazil setup would be welcome.**

Interactive
comment

[AUTHORS RESPONSE 10]: This is an important remark! Checking the estimates again and looking for an explanation it turned out that our initial analysis was incomplete, ignoring some model internal dynamics. A deeper look revealed that the global deforestation estimate in the Brazil setup is actually unreliable as it is caused by unrealistic production shifts within the Rest of the world (ROW) region. We corrected that in the revised version and added some discussion for the figure:

*[TEXT EDIT 10] "Comparison with historical data sets as well as projections on forest cover show that the differences between mappings are rather small compared to the overall uncertainty in these numbers. Nevertheless, a deeper look into the simulations uncovers that the global numbers of the Brazil-centric setup are unreliable as the reduced deforestation rate compared to the default setup is a consequence of the applied mapping. As the ROW region basically acts as a huge free trade region it can fulfill strong demand pressure coming from Sub-Saharan Africa with production from elsewhere, while trade limitations in the default setup limit this exchange and trigger deforestation within Sub-Saharan Africa Africa (Dietrich, 2018, compare m4p_default_validation.pdf p1558 and m4p_brazil_validation.pdf p1465). In the case of LAM both runs show a rather similar picture in the aggregated forest cover projections for the region and it is not possible to clearly reject one of them. This is particular important as the regional aggregates in LAM are in the scope of both mappings and therefore should be sound. When choosing between them, one has to decide whether spatial details in Brazil or global trade patterns are the more decisive factor for accurate estimates of regional forest cover in LAM."*

**[REFEREE COMMENT 11]: Correction suggestions for spelling/syntax/punctuation:**

- **page1_line5 computationally intensive**
- **page3_line27 region-specific**

[Figure]

- **page5_line1 choose a regional aggregation, with the country level () as the highest....**

- **page5_lines45 food-demand**

- **page8_line28 realizations**

- **page9_line25 The model outcomes at the cluster level**

- **page9_line27 data pre-processing at ISO country or 0.5 degree level**

- **page13_line4 less -> fewer**

- **page16_line11 and other research institutions, as enabled by**

[AUTHORS RESPONSE 11]: We corrected the text as suggested.

---

## Author Comment (AC2) · 20 Feb 2019

We would like to thank the Referee for all the efforts and the valuable remarks which helped to considerably improve the paper.

**[REFEREE COMMENT 1]: The paper presents the IT architecture of the MAgPIE framework focusing on two features: modularity and flexibility of spatial resolution. This is a rather technical paper which is well written and easily understandable in spite of its technicity. This type of paper is welcomed to improve the transparency of models and help interpreting their result. Here are my comments: 1. The presentation of the modules (p. 5-7) raise a number of issues: (i) The definition of the modules is sometimes vague. The "costs" module is not**
**easy to grasp: what kind of aggregates does it make? In fact, we wonder why this is a separate module for it. Why is the aggregation not done in the corresponding module? The "production" module is defined as aggregating cellular production to the regional level, but how the cellular production is defined? I would say this result from the "yields" and "crop" module, but this is not clear from the text and from Figure 1.**

[AUTHORS RESPONSE 1]: In the revised manuscript we try to be more precise. In the case of the "costs" module we talk now about "summation" instead of "aggregation" specifically stating that the outcome is total costs to make it more transparent that it is not more than a total sum of costs. We changed the formulation to

*[TEXT EDIT 1] "Calculates total costs by summing up all costs in the model including production costs, investments into research and development or land expansion, tax expenditures, and mitigation costs."*

[AUTHORS RESPONSE 1]: In case of module "production" we realized that the previous formulation was suggesting that crop production is calculated in the production module, which would actually be in contradiction to the visualization in Figure 1. In fact, crop production is already calculated in the crop module and just merged with other production information in the production module. To make this transparent we updated the descriptions of modules "production" and "crop" accordingly. We also added a sentence explaining for what purpose production values are aggregated to regional production in the production module. More specific questions such as the question how cellular crop production is defined are answered in the model documentation in the supplementary material and therefore not discussed here. New description of the "production" module is now:

*[TEXT EDIT 1] "Merges production values including crop-based production and livestock-based production into one production variable. Aggregates cellular production to the regional level for modules only interested in regional production levels."*

[Figure]

New description of the "crop" module is now:

*[TEXT EDIT 1] "Simulates crop production and competition of different crop types for cropland, accounting also for crop rotation requirements. Estimates the terrestrial carbon pools of croplands."*

The updated formulation is now in line with the implementation in the model as well as the visualization in figure 1.

**[REFEREE COMMENT 2]: (ii) Prices are almost absent from the picture while they are a key element of the system. They are the primary drivers of the intensification mechanisms which are for this reason unclear here: is there some livestock intensification? Does the technological change react to price or it is exogenous? Also how is the fertilizer use treated? In so doing we don't see the substitution possibilities between production factor while this is basically what the model represent.**

[AUTHORS RESPONSE 2]: In MAgPIE prices are implicitly modelled as marginals of the model constraints and are therefore not part of the module descriptions. Intensification as well as all other decisions in the module are coming from an interplay of physical constraints and costs associated to activities in the model. We explain this now in the paper with the following sentence added to the brief history of MAgPIE section:

*[TEXT EDIT 2] "Prices are implicitly modeled as marginals of the model constraints. Intensification as well as other decisions in the model are coming from an interplay of physical constraints and costs associated to activities in the model."*

[AUTHORS RESPONSE 2]: The implemented intensification processes are explained in the model documentation. As explained in the description of the tc module, the available TC implementation is endogenous and intensification can be triggered via investments into technological change. Pasture yield intensification in MAgPIE 4.0 is available in 2 realizations (see yields module): In the realization "biocorrect" pasture

yields grow parallel to crop yields based on TC investments, in "dynamic_aug18" pasture intensification is based on exogenous scenario assumptions. Besides pasture intensification there is also a scenario based conversion of livestock production systems implemented which is explained in detail in the model description of the livestock module. As many aspects are answered in the model documentation we make sure that it is more noticeable in the paper. Especially, we added a link to the documentation in the description of Figure 1 (model linkages).

**[REFEREE COMMENT 3]: (iii) Finally some feedback loops seem to be lacking, e.g.: the production of residues should affect the bioenergy module; the crop module should affect the livestock through feed production; the livestock production may affect the yield through manure and the availability of land may have an impact on yields.**

[AUTHORS RESPONSE 3]: A feedback of residue production on bioenergy demand is indeed not considered in the current setup. Instead the framework currently works with external demand scenarios for residues which are chosen consistently to the demand calculations in the bioenergy module. We are currently working on improving the residue implementation and expect changes for the next MAgPIE releases. The connection between livestock and crop production actually exists in two ways which are implied in Figure 1. The first connection goes via feed demand which triggers additional crop production (Livestock -> Demand -> Trade -> Production -> Crop). The second one goes through livestock triggering pasture demand which is competing with the crop module for land (Livestock -> Pasture -> Land -> Crop). Yield increases through manure are not explicitly modeled but land availability is affecting yields as expected by the reviewer. All these details about the interactions are explained in the model documentation and therefore not part of the paper. As mentioned above, we link the model documentation more noticeable in the paper. Furthermore, we explain now in the text at the end of subsection "Modules" that also modules not directly linked are connected to each other via other modules:

*[TEXT EDIT 3] "If modules are not directly linked it does not mean that they do not interact with each other. In some cases the feedback loops go through a combination of modules rather than being direct links. An example is the livestock module which is triggering feed demand in the demand module, which is, via trade and production module, triggering production in the crop module."*

**[REFEREE COMMENT 4]: The last two points reveal the difficulty of representing a system in a modular way, as each module strongly interacts with the other, making the frontier between them sometimes meaningless. Livestock and crop production system are typical examples as they are generally strongly integrated. This point is an important barrier to the modular representation which should be discuss in depth and better justified. In some cases the modular representation is appropriate for modeling reality, however in some cases it could put into question the consistency of the model. Most importantly, this approach may be seen as only compatible with conventional agriculture, and not with alternative agricultural systems promoting a systemic approach.**

[AUTHORS RESPONSE 4]: This is a very important remark which requires some discussion. Our experience so far showed the general applicability and usefulness of a modular structure is also quite high for strongly integrated systems. One aspect which we avoided to discuss so far, but is relevant in this context is the question of persistence of modular structures. Having a completely static modular structure would indeed significantly limit the modeling capabilities of MAgPIE. What we are dealing with instead is a semi-static structure, in which module definitions are valid on a longer time scale than their underlying realizations, but are also allowed to change from time to time. Introducing or removing interfaces is possible as well as creating, splitting, merging or deleting of modules. Doing so allows to adapt the framework structure to new challenges which cannot be reflected in the given structure. Concerning module separations in closely linked systems our experience is that this is useful as well and helps to understand the interactions better. The difference between closely and loosely linked

modules is usually just the number of interfaces and interactions. What is gained by the modular structure is a better control over interactions in the model as only specifically designed interactions are possible, while all other interactions will be detected as a code violations. In the early stages of modularizing MAgPIE this uncovered for instance cross-links in MAgPIE in which completely unrelated variables were used as surrogates for processes not covered by the model. While this might be desirable in some cases it can easily lead to unrealistic model behavior when users are not aware of such a link and by default do not expect that such a link exists. A modular structure does not generally prohibit such links but it makes them clearly visible and forces the researcher to think about it. We extended the main text by adding a discussion addressing this issue:

*[TEXT EDIT 4] "One main improvement introduced in MAgPIE 4 is the full code modularization. It is used as a tool to make the model better manageable as it structures the code in self-containing components which are interacting via interfaces with each other. It makes existing and missing interactions in the model better visible and allows to easily replace components by alternative implementations. While the modular structure is rather intuitive for a system with loosely linked components one could argue that it might prevent a proper implementation of strongly integrated systems. Our experience is that, while the modular concept is working best for clearly separable systems, it also works in all other cases. The difference with strongly integrated systems is that the amount of interfaces and the required effort for developing new realizations are higher. Nevertheless, it still improves transparency in terms of model interactions and does not exclude any systems or dynamics from being represented in the model. Modules are also not static and the modular structure itself can and will also be changed if required. Modules might get created, deleted, merged or split over time. Module interfaces might get extended, reduced or modified. As both happens less frequently than changes within modules the modular structure can be best described as semi-static."*

**[REFEREE COMMENT 5]: 2. The key evaluation examples are not very informa-**

**tive in the context of this paper. The paper presents the "framework" and not really the model (i.e., the economic and biophysical mechanisms represented). For this reason, having, e.g., an evaluation of the crop yield simulated by the model is not very relevant to the paper. We would rather expect an evaluation of the modular architecture, as the authors did for the spatial resolution in the next section. How the modules perform running together vs standalone?**

[AUTHORS RESPONSE 5]: We agree that the key evaluation examples are a bit unrelated to the topic of the paper. We like the suggestion to show module-related examples instead. Consequently, we moved the key evaluation example to the appendix and added a new section "Impact of module realizations" to the model outputs section. The new section discusses three different applications of the modular structure (two cases in which alternative realizations are used and one case in which a module is run standalone). The new text reads as follows:

[revised manuscript text omitted]

**[REFEREE COMMENT 6]: 3. The evaluation of the spatial flexibility is interesting however I did not really understand why there is so much difference in the default and Brazil-specific settings. Why is there much specialization in the default setting? And to what extent does it affect the spatial deforestation/reforestation pattern in Brazil? Also, the paragraph beginning on p13 l4 is quite difficult to understand (why is there 200 clusters in the default version and 500 in the region-specific one ?)**

[AUTHORS RESPONSE 6]: We reformulated the paragraph previously beginning on p13 l4 to make it easier to understand and to better explain the increase of clusters

from 200 to 500:

[TEXT EDIT 6] *"Figure 4 shows a setup with a specific focus on Brazil. To gain higher spatial detail in Brazil it comes with a higher number of clusters in total. Brazil (BRA) is simulated as a world region together with its most important trade partners (Rest of Latin America (LAM), United States (USA), China (CHA) and Europe (EUR)). Remaining countries, less relevant for a Brazil-centric study, are merged to a single region (ROW). Furthermore, the cluster allocation of 500 clusters in total has been shifted in favor of Brazil: Roughly four times more clusters are allocated to Brazil (306) compared to a default distribution of clusters. At the same time the rest of the world region receives only roughly 0.7 times the number of clusters it would usually get (37), leaving room for a balanced number of clusters for all other regions. Detail gained for Brazil is attained with reduced detail for the rest of the world to keep the model complexity manageable for the applied solver."*

[AUTHORS RESPONSE 6]: To explain the observed specialization in the spatial patterns we added another section to the discussion addressing this issue:

[TEXT EDIT 6] *"The observed specialization is a consequence of the homogeneous biophysical characteristics within each cluster which lead to either-or-decisions in the model. It will either fully take a cluster into production or ignore it completely. In the default setup this effect is very pronounced due to the low number of clusters within Latin America. With more clusters, as in the Brazil setup, clusters better grasp the real spatial distributions of biophysical characteristics in the region and therefore lead to a more diverse picture. Whereas this effect is especially relevant for regional studies with focus on spatial patterns, it is less critical for global dynamics as long as the spatial aggregation is not introducing any systematic biases to the model."*

[AUTHORS RESPONSE 6]: Furthermore, we improved the discussion of forest cover development in Latin America and globally under both setups:

[TEXT EDIT 6] *"Comparison with historical data sets as well as projections on for-*

*est cover show that the differences between mappings are rather small compared to the overall uncertainty in these numbers. Nevertheless, a deeper look into the simulations uncovers that the global numbers of the Brazil-centric setup are unreliable as the reduced deforestation rate compared to the default setup is a consequence of the applied mapping. As the ROW region basically acts as a huge free trade region it can fulfill strong demand pressure coming from Sub-Saharan Africa with production from elsewhere, while trade limitations in the default setup limit this exchange and trigger deforestation within Sub-Saharan Africa Africa (Dietrich, 2018, compare m4p_default_validation.pdf p1558 and m4p_brazil_validation.pdf p1465). In the case of LAM both runs show a rather similar picture in the aggregated forest cover projections for the region and it is not possible to clearly reject one of them. This is particular important as the regional aggregates in LAM are in the scope of both mappings and therefore should be sound. When choosing between them, one has to decide whether spatial details in Brazil or global trade patterns are the more decisive factor for accurate estimates of regional forest cover in LAM."*

———————————————————

[Figure]

**Drivers**

Population   GDP   Scenarios

**Policies & Climate Action**

GHG
**Greenhouse Gas Policy**

**Carbon**

**Nitrogen**

**Marginal Abatement Cost Curves**

**Methane**

**Landconversion**

**Soil Organic Matter**

**Food**

**Costs**

**Forestry**

**Water**

**Material**

**Interest rate**   **Optimization**

**Demand**

**Pasture**

**Land**

**Bionergy**

**Yields**

**Urban**

**Technological Change**

**Trade**

**Livestock**

**Natural Vegetation**

**Factor costs**

**Crop**

**Processing**

**Production**

**Residues**

**Transport**

**Socio-Economic Data**

**Biogeophysical & Climate Impact Data**

**Fig. 1.**

[Figure]

**Fig. 2.**

---

## Author Comment (AC3) · 20 Feb 2019

We would like to thank the referee for the time spent on reviewing our paper and the valuable remarks which pointed out some important issues and helped to further improve the paper.

**[REFEREE COMMENT 1] General Comments This paper describes the model MAgPIE 4, its history and release as an open source modelling framework. It serves as an introduction to the modelling framework as it is released, and as such it is well-suited as a citation for future users. There is good explanation of the structure of the model, including its modularity and references to further documentation. However, the effects of running modules under different setups**

[Figure]

is not explored. The article also presents a case study with a focus on Brazil that demonstrates the regional flexibility of the framework and how different regional definitions may affect results. This last topic is particularly interesting and deserves to be expanded a bit. In particular what may be behind the different results that arise from changing regional definitions. Finally, the absence of any description of the objective function of the model framework is an omission that should be corrected. Therefore, I recommend that this article be accepted with minor revisions as detailed below.

[AUTHORS RESPONSE 1] Please see our answers below.

**[REFEREE COMMENT 2] There is much emphasis on the modularity structure of the framework, but the article does not go beyond description of this modularity. It lacks any mention of how choices of modular setup may affect results. For example, how does a module behave in standalone versus integrated mode? Or how does changing a specific module realization affect other modules? This analysis should be included here as, for example, a case study of one specific module. Even if this is performed in a separate article or online documentation resource, a quick summary of the results of such an experiment should be included for illustrative purposes.**

[AUTHORS RESPONSE 2] We agree with the referee that detail was missing here. To address this issue we moved the discussion of key evaluation examples to the appendix and replaced in with a comparison of three applications of the modularity approach (2 cases in which alternative realizations were used and 1 standalone case). The new text reads as follows:

[revised manuscript text omitted]

*expansion of cropland and reduction in forest areas."*

**[REFEREE COMMENT 3] The optimization methodology should be better explained. There is only a brief mention of the method in the description of the optimization module, which states that the model "minimizes total system cost" (p7, l14). How is this done? Is the model dynamic recursive? This has been explained in other articles using MAgPIE, but it should be included here, either in the main text or the appendices. A description of the objective function and the optimization method is in order.**

[AUTHORS RESPONSE 3] This is indeed a critical omission. We extended the text in various locations with information about the optimization methodology (which is indeed dynamic recursive cost minimization). Details which go beyond that can be found in the referenced model documentation. In detail we added it to the first sentence in the MAgPIE history:

*[TEXT EDIT 3] "MAgPIE was first introduced in Lotze-Campen et al. (2008) as recursive dynamic cost minimization model, simulating crop production, land-use patterns, and water use for irrigation in a spatial resolution of three by three degrees and interregional trade between 10 world regions."*

[AUTHORS RESPONSE 3] As the recursive dynamic logic is part of the inner layer of the framework we added it to its description in the framework architecture section:

*[TEXT EDIT 3] "The inner layer written in GAMS (GAMS Development Corporation, 2016) contains the optimization model with all its equations and constraints, the recursive dynamic logic which triggers the optimization for each time step consecutively and forwards results to the next time step and the code modularity implementation."*

[AUTHORS RESPONSE 3] Additionally, we extended the description of the optimization module to:

*[TEXT EDIT 3] "Minimizes total costs of the optimization problem for each time step*

*using different optimization strategies to reduce run time."*

**[REFEREE COMMENT 4] Another important issue is the expansion of the discussion on what drives the changes in results from different regional aggregation. In particular, the difference in global forest cover changes by about 10% when using the Brazil setup should be explored in more detail. Even if it is simply the result of coarser resolution in the ROW region, it would be interesting to hear more about the interpretation of these results. Is 10% an acceptable uncertainty level? Which global regions are most affected by the changing the regional definitions? Why? This would expand the discussion section as well.**

[AUTHORS RESPONSE 4] Discussion of this part came indeed a bit short. Looking more into it we found that the global forest cover numbers in the Brazil setup should not be used as they arise from unrealistic production shifts within the ROW region. We added a paragraph to the discussion specifically addressing this issue:

*[TEXT EDIT 4] "Comparison with historical data sets as well as projections on forest cover show that the differences between mappings are rather small compared to the overall uncertainty in these numbers. Nevertheless, a deeper look into the simulations uncovers that the global numbers of the Brazil-centric setup are unreliable as the reduced deforestation rate compared to the default setup is a consequence of the applied mapping. As the ROW region basically acts as a huge free trade region it can fulfill strong demand pressure coming from Sub-Saharan Africa with production from elsewhere, while trade limitations in the default setup limit this exchange and trigger deforestation within Sub-Saharan Africa Africa (Dietrich, 2018, compare m4p_default_validation.pdf p1558 and m4p_brazil_validation.pdf p1465). In the case of LAM both runs show a rather similar picture in the aggregated forest cover projections for the region and it is not possible to clearly reject one of them. This is particular important as the regional aggregates in LAM are in the scope of both mappings and therefore should be sound. When choosing between them, one has to decide whether spatial details in Brazil or global trade patterns are the more decisive factor for accurate*

*estimates of regional forest cover in LAM."*

[AUTHORS RESPONSE 4] Furthermore, we added a section discussing the specialization observed in the spatial patterns, where it comes from and how it affects global dynamics:

*[TEXT EDIT 4] "The observed specialization is a consequence of the homogeneous biophysical characteristics within each cluster which lead to either-or-decisions in the model. It will either fully take a cluster into production or ignore it completely. In the default setup this effect is very pronounced due to the low number of clusters within Latin America. With more clusters, as in the Brazil setup, clusters better grasp the real spatial distributions of biophysical characteristics in the region and therefore lead to a more diverse picture. Whereas this effect is especially relevant for regional studies with focus on spatial patterns, it is less critical for global dynamics as long as the spatial aggregation is not introducing any systematic biases to the model."*

**[REFEREE COMMENT 5] Also, the initial land cover map most likely plays an important role on the future projections. As such, it would be advisable to include a figure with the base year land cover map, even though this may be extracted from Hurtt et al. (2018). In fact, land cover maps for other milestone years the authors deem important may also help the user to understand model dynamics.**

[AUTHORS RESPONSE 5] We agree with the referee that initial land cover is a crucial input which is affecting future projections. However, due to the nature of the paper focusing on the MAgPIE framework we felt that the benefit of showing land cover maps in the paper would be limited. For interested readers we added a sentence to the SSP results discussion (Appendix A2) mentioning that further details (including the land cover maps in NetCDF format) can be found in the uploaded supplementary material, including the land cover maps of the corresponding runs:

*[TEXT EDIT 5] "More information information about the runs can be found in the corresponding evaluation documents (Dietrich, 2019b) and model runs (Dietrich, 2019a).*

[Figure]

*The latter contains for instance NetCDF-files with spatial land cover information of the corresponding runs (cell.land_0.5.nc)."*

**[REFEREE COMMENT 6] Technical Comments P13 l4: repeated "with"**

[AUTHORS RESPONSE 6] We deleted the duplicate "with"

———————————————————

[Figure]

**Fig. 1.**

---

## Author Response (AR1)

Dear Editor,

    on behalf of all co-authors I hereby send you the marked-up manuscript version showing all changes between the previous and current submission. The corresponding authors responses are uploaded in the interactive discussions of the manuscript at

5  https://doi.org/10.5194/gmd-2018-295.

Kind regards,
Jan Dietrich

[revised manuscript text omitted]